# MicroRNAs Involved in Carcinogenesis, Prognosis, Therapeutic Resistance, and Applications in Human Triple-Negative Breast Cancer

**DOI:** 10.3390/cells8121492

**Published:** 2019-11-22

**Authors:** Lei Ding, Huan Gu, Xianhui Xiong, Hongshun Ao, Jiaqi Cao, Wen Lin, Min Yu, Jie Lin, Qinghua Cui

**Affiliations:** 1Lab of Biochemistry & Molecular Biology, School of Life Sciences, Yunnan University, Kunming 650091, China; dingleiynu@yun.edu.cn (L.D.); albertwesker0112@163.com (H.G.); xiongxianhui1995@163.com (X.X.); aohongshun@163.com (H.A.); 18487162936@163.com (J.C.); linwen967@163.com (W.L.); yumin@ynu.edu.cn (M.Y.); linjie@ynu.edu.cn (J.L.); 2Key Lab of Molecular Cancer Biology, Yunnan Education Department, Kunming 650091, China

**Keywords:** triple-negative breast cancer (TNBC), miRNAs, tumorigenesis, prognosis, chemotherapy and radiotherapy resistance, therapeutic strategies, epigenetic mechanisms

## Abstract

Triple-negative breast cancer (TNBC) is the most aggressive, prevalent, and distinct subtype of breast cancer characterized by high recurrence rates and poor clinical prognosis, devoid of both predictive markers and potential therapeutic targets. MicroRNAs (miRNA/miR) are a family of small, endogenous, non-coding, single-stranded regulatory RNAs that bind to the 3′-untranslated region (3′-UTR) complementary sequences and downregulate the translation of target mRNAs as post-transcriptional regulators. Dysregulation miRNAs are involved in broad spectrum cellular processes of TNBC, exerting their function as oncogenes or tumor suppressors depending on their cellular target involved in tumor initiation, promotion, malignant conversion, and metastasis. In this review, we emphasize on masses of miRNAs that act as oncogenes or tumor suppressors involved in epithelial–mesenchymal transition (EMT), maintenance of stemness, tumor invasion and metastasis, cell proliferation, and apoptosis. We also discuss miRNAs as the targets or as the regulators of dysregulation epigenetic modulation in the carcinogenesis process of TNBC. Furthermore, we show that miRNAs used as potential classification, prognostic, chemotherapy and radiotherapy resistance markers in TNBC. Finally, we present the perspective on miRNA therapeutics with mimics or antagonists, and focus on the challenges of miRNA therapy. This study offers an insight into the role of miRNA in pathology progression of TNBC.

## 1. Introduction

Over the past decades, with the continuous advances in early detection, the development of personalized therapy, the improvements in chemotherapy, the survival rates of patients with breast cancer have remarkably increased. However, breast cancer is still the leading cause of cancer mortality for women worldwide. Breast cancer is recognized as a heterogeneous disease, involving multiple oncogenic biological pathways and/or genetic alterations. According to the comprehensive gene expression profiling, breast cancer can be categorized into five major subtypes: Luminal A (estrogen receptor (ER) and/or progesterone receptor (PR) positive, and human epidermal growth factor receptor 2 (HER2) negative), luminal B (ER and/or PR positive and HER2 positive), HER2 enriched (ER negative, PR negative, and HER2 positive), basal-like (ER negative, PR negative, HER2 negative, cytokeratin 5/6 positive, and/or epidermal growth factor receptor (EGFR) positive), and normal breast-like cancers [1]. Basal-like breast cancer makes up about 15–20% of breast cancers and has drawn much attention due to short relapse-free and low survival rate [2]. Many studies have indicated that basal-like breast cancer type shares many overlapping features with triple-negative breast cancer (TNBC) that is defined by the lack expression of ER, PR, and HER2 receptor, and characterized by early relapse, aggressive tumor growth, unresponsiveness to treatment, distant recurrence, and lowest survival rate. TNBC accounts for approximately 15–25% of breast cancer diagnoses with poor outcome by both antiestrogen hormonal therapies and monoclonal antibody-based therapies, which are targeted efficiently for non-TNBC patients. Currently, cytotoxic chemotherapy and radiotherapy remain the approved treatment for TNBC patients in the early or advanced stages [3]. Thus, identification of the novel molecular markers that target the growth and carcinogenesis of TNBC cells is an urgent clinical need to improve the diagnosis and therapies for patients.

The epigenetic alterations and microRNA (miRNA) dysregulation are known to be important in silencing of gene expression implicated in TNBC, and either suppress or activate multiple genes at the pre- and post-transcriptional levels, respectively. MiRNAs are endogenous, approximately 19–25 nucleotides non-coding RNAs, and negatively regulate gene expression of specific mRNA targets. The majority of miRNAs are located in endonuclear noncoding regions, such as introns of protein-coding genes; however, miRNAs were also observed in exons of genes. Masses of known human miRNA are encoded in fragile chromosomal regions which are sensitive to amplification, deletion, or translocation during the occurrence and development of cancer [4]. The precursor miRNAs involve a complex process in the nucleus, and are then exported into cytoplasm to further process to become mature miRNAs (Figure 1A). Briefly, miRNAs are transcribed from different genomic locations by RNA polymerase II enzyme (pol II) as a long primary transcript (pri-miRNAs) and cleaved by Drosha (RNase III family) and its cofactor DiGeorge syndrome critical region in gene 8 (DGCR8) to yield the precursor miRNA (pre-miRNA) in the nucleus. After transfer into the cytoplasm by Exportin-5 (XPO5) in a Ran-GTP-dependent manner, the pre-miRNA is further processed into miRNA:miRNA* duplex by Dicer in concert with trans-activation response RNA-binding protein (TRBP) or protein activator of interferon-induced protein kinase EIF2AK2 (PACT, also known as PRKRA) cofactors [5]. Then, the duplex unwinds and the mature single-stand miRNA is subsequently incorporated into the RNA-induced silencing complex (RISC) to form a miRNA-induced silencing complex (miRISC) with Argonaute (Ago) family proteins [6]. As shown in Figure 1B, the miRISC complex pairs with its complimentary target recognition on mRNA in a perfect or an imperfect manner in the 3′-untranslated region (3′-UTR), thereby, it silences the expression of the target mRNA either by mRNA cleavage or by translational repression [7,8,9,10,11]. Recently, some researchers have indicated that miRNAs can upregulate, rather than repress, the translation of their target mRNA, but this is the minority [12]. In addition, the epigenetic alterations are potentially reversible in neoplasia, which offers a new opportunity for cancer clinical management in TNBC, since it is different from mutation. The epigenetic alteration is centered on the modification of CpG islands, histones, and nucleosome positioning. Especially, the aberrant DNA methylation associated with the addition of a methyl group to a cytosine base in the dinucleotide sequence CpG islands in DNA promoter region, which generally correlates with a silenced gene. Moreover, the histones can undergo numerous modifications, such as lysine acetylation, lysine and arginine methylation, serine threonine and tyrosine phosphorylation, and lysine ubiquitination. DNA, histone-modifying enzymes, and various miRNAs are involved in epigenetic modification [13].

Each miRNA has multiple targets, while the same gene can also be influenced by several miRNAs. To date, there are more than 900 identified human miRNAs [14] transcribed either as individual units, polycistronic clusters, or in concert with the protein-coding host gene [15]. The dysregulation of miRNAs are involved in different cancer pathogenesis of solid tumors, such as apoptosis, inflammation, stress response, cell cycle, proliferation; the tumor microenvironment facilitates tumor development, differentiation, morphogenesis, progression and metastasis, and acted either in dominant or recessive roles, including breast cancer [16,17,18]. Emerging in vitro and in vivo studies open up a new area that miRNAs have the potential to serve as promising biomarkers for diagnosis and the development of novel anticancer drugs [19]. In this review, we attempt to summarize the potential clinical value and the roles of miRNAs involved in the pathogenesis of TNBC, such as the regulation of epigenetic mechanisms, epithelial-to-mesenchymal transition (EMT), maintenance of stemness, proliferation, metastasis, and apoptosis. We also summarize their roles in classification, prognostic and therapeutic resistance in TNBC, the potential use of miRNAs as relevant drug targets, possible therapeutic candidates, and the challenges.

## 2. The Mechanisms Altering miRNA Expression in TNBC

Many studies have shown that the dysregulation of miRNA expression is a rule rather than the exception in TNBC. The tight integration of miRNAs in physiological or pathological regulation circuits has maybe become a problem because the aberrant expression or deletion of a small number of miRNAs reduces the control of a large number of target genes’ expression in cells. Several mechanisms influence the miRNA expression during the development of TNBC, and as a consequence, the tumors often present altered expression levels of mature miRNAs.

### 2.1. The Genetic Alteration of miRNA Expression in TNBC

Genomic and epigenetic alteration usually leads to the miRNA deregulation, which is often involved in chromosomal abnormalities, associated with the deletion, amplification, or translocation of miRNAs, and potentially resulting in initiation and development of cancer [20]. The epigenetic covalent modifications usually physically influence the density of the chromatin fiber, leading to determine whether chromatin is in the accessible status (euchromatin), or in the inaccessible one (heterochromatin) [13]. Calin et al. showed that about more than half (98 of 186) of the annotated human miRNA genes are located in cancer-associated genomic fragile regions, as well as the areas of the genome in the minimal regions of loss of heterozygosity, minimal regions of amplification (minimal amplicons), or common breakpoint and rearrangement regions [21]. Many miRNA family locis emerge to be in close proximity with other miRNAs involved in tumor development. Hence, they are transcribed as polycistronic messages in a single transcript unit or overlapped in the host transcripts [22]. For example, several let-7 family members, together with miR-125b-1, miR-100, and miR-34a-1/2, were found to be located at fragile sites of human chromosomes (11q23-q24-D) associated with the aberrant miRNA expression in breast cancer [21]. Cascione et al. revealed that the cluster miR-17/92 and miR-200 family were upregulated, while two members of the let-7 family (let-7b and let-7c) were downregulated from the 116 deregulated miRNAs in the first set of primary TNBC and normal tissues [23]. Chang et al. documented that seven polycistronic miRNA clusters preferentially dysregulated expressed in TNBC, and two of them (miR-143/145 at 5q32 and miR-497/195 at 17p13.1) were markedly downregulated, while the other five miRNA clusters (miR-17/92 at 13q31.3, miR-182/183 at 7q32.2, miR-200/429 at 1p36.33, miR-301b/130b at 22q11.21, and miR-532/502 at Xp11.23) were markedly upregulated in TNBC by hierarchical clustering analysis [24]. Cantini and co-workers obtained that miR-199/214 and miR-532/502 clusters promote the TNBC phenotype through their control of proliferation and EMT [25].

### 2.2. Defects of Genes in the miRNA Biogenesis Pathway

In addition to genomic and epigenetic alterations, the defects of the genes in the miRNA biogenesis pathway globally alter the miRNA expression and make the cell suitable for oncogenic changes. For example, reduced Dicer and Drosha expression is associated with higher-grade breast cancer and shorter metastasis-free survival or disease-free survival [26,27]; especially, the reduced expression of Dicer is involved in TNBC phenotype [28]. In breast cancer, the nucleolin (NCL) protein directly interacts with the DGCR8 and Drosha microprocessor complex in the nucleus to affect the biogenesis of miR-15a/16 at the primary to precursor stage of processing [29]. Pichiorri et al. indicated that the NCL, an integral component of the Drosha/DGCR8 microprocessor complex, post-transcriptionally regulated the maturation of a set of metastasis-promoting miRNAs’ expression (miR-21, miR-221/222 cluster, and miR-103), involved in breast cancer initiation, progression, and drug resistance [30]. Furthermore, the high expression of miRNA biogenesis gene XPO5, a key protein responsible for exporting pre-miRNAs through the nuclear membrane to the cytoplasm, affects the risk of breast tumorigenesis [31]. Lin et al. showed that the high expression of the cytoplasm trans-activation-responsive-RNA-binding protein 2 (TARBP2), the Dicer stabilizing protein, had poorer disease-free survival, overall survival, and unfavorable prognosis in TNBC [32].

On the other hand, given that a great deal of transcription factors play roles in miRNAs’ fundamental biological processes, it is not surprising that many tumor-related genes function as transcription factors that mainly influence the expression levels of miRNAs by activating the transcription of pri-miRNAs. Many evidences indicate that the miRNA biogenesis pathway and transcription factors work cooperatively to form as a functional feedback loop, where the miRNA expression levels can influence the expression of transcription factors and vice versa [33,34]. For example, the drosophila mothers against decapentaplegic protein (SMAD), the p53 protein family, ataxia telangiectasia mutated (ATM), c-Myc, and E2F are further discussed [22,35]. In breast, tumor suppressor breast cancer 1 (BRCA1) participated in the assembly of the Drosha microprocessor complex, as well as interacted with the transcription factor of Smad3, p53, and DExH-box helicase 9 (DHX9) RNA helicase to regulate the processing of the pri-miRNAs let-7a-1, miR-16-1, miR-145, and miR-34a, which indicated that BRCA1 played critical roles in miRNA biogenesis and maintenance of genomic stability [36]. Furthermore, the hypoxic transcription factor of EGFR suppresses specific tumor suppressor-like miRNA maturation through phosphorylation of AGO2 involved in the regulation of the RISC [37], thus enhancing cell survival and invasiveness and correlating with poorer overall survival in breast cancer patients.

## 3. miRNA Dysregulation in TNBC

MiRNAs are found stable in whole blood, plasma, and serum samples and are easily detected; they have been recognized as potential diagnostic, predictive, and prognostic biomarkers in TNBC patients [38]. The expression levels of miRNA can help to separate breast cancer specimens from normal tissues, and accurately distinguish TNBC from non-TNBC types.

### 3.1. Classification of TNBC from Other Breast Cancer Subtypes with miRNA Signatures

Approximately 75% of TNBC express basal markers; consequently, TNBC is frequently and inaccurately taken as the surrogate maker for the basal-like sub-type breast cancer [39]. TNBC is connected with higher nuclear grade, higher histological grade, and higher genomic instability. In TNBC patients, the miR-17-92 cluster shows as remarkably upregulated, exhibiting a modest change for a robust diagnostic marker for the districting from other tumor subtypes [40]. Savad et al. analyzed the expression of miR-21, miR-205, and miR-342 in 59 patients with breast cancer, and showed that both miR-205 and miR-342 expressions were significantly downregulated in the TNBC group compared with other breast cancer subtypes, suggesting that miR-205 and miR-342 can be used as potential biomarkers for the diagnosis of TNBC [41]. Moreover, miR-342 showed upregulated expression in ER-positive and HER2/neu-positive luminal B breast tumors, and showed a downregulation in TNBC [38]. Gasparini et al. identified a panel of 4-miRNA signature given by miR-155, miR-493, miR-30e, and miR-27a expression levels, which allowed subdivision of TNBC into core basal (CB) or five negative (5NP) subgroups; CB subgroups tend to predict poor outcomes compared to 5NP [42]. Calvano Filho et al. indicated that there was enhanced expression of miR-17-5p, miR-18a-5p, and miR-20a-5 in TNBC compared with luminal A breast cancer subtype [43]. Additionally, Thakur et al. demonstrated that a panel of three oncogenic miRNAs (miR-21, miR-221, miR-210) and the tumor suppressor miRNA (let-7a) showed significant overexpression, while two tumor suppressor miRNAs (miR-195 and miR-145) were downregulated in TNBC patients compared with triple positive breast cancer (TPBC) patients. It is suggested that this panel of six miRNA signatures may serve as a potential non-invasive biomarker for an early and reliable diagnosis of TNBC patients [44]. Altogether, these findings elucidated these dysregulated miRNAs as candidate diagnostic biomarkers in TNBC (Table 1).

### 3.2. Classification of TNBC from Non-TNBC with miRNAs Signatures

There have been multiple studies demonstrating the dysregulated expressions of miRNA associated with TNBC, attempting to reflect the histopathological features of the tumor via the association of miRNA profiling with genomic classes (Table 1). The expression of miR-21, miR-146a and miR-182 was significantly overexpressed in TNBC. MiR-10b, miR-21, and miR-182 were significantly associated with lymph node metastases occurrence in TNBC, and miR-10b was associated with grade III occurrence [45]. Global miRNA expression profiling in the serum of TNBC patients showed that miR-190a, miR-136-5p, miR-126-5p, miR-135b-5p, and miR-182-5p were downregulated relative to the healthy, associating with the development and progression of TNBC [46]. Yang et al. summarized three independent studies and found that six miRNAs (miR-146a, miR-100, miR-125b, miR-29a, miR-222, and miR-221) are upregulated, while five miRNAs (miR-200a, miR-200c, miR-141, miR-375, and miR-203) are downregulated in TNBC compared to non-TNBC. They also found that for twelve miRNAs, four (miR-138-5p, miR-4324, miR-4800-3p, and miR-6836-3p) were upregulated, and the other eight (miR-363-5p, miR-182-5p, miR-141-3p, miR-339-5p, miR-4655-3p, miR-4784, miR-664b-5p, and miR-6787-5p) were downregulated in TNBC cell lines (MDA-MB-231 and Hs578T) compared to non-TNBC cell lines (MCF-7 and SK-BR-3) [47]. In a more recent study, Paszek et al. unraveled the expression of miR-190a, miR-136-5p, and miR-126-5p, showing that they were significantly reduced, whereas the expression of miR-135b-5p and miR-182-5p was significantly increased, and there were linear correlations between the tumor size and the expression levels of miR-126-5p and miR-135b-5p in TNBC patients compared to normal breast tissues [19]. The miRNA expression profiles of the human TNBC cell line were evaluated, and 107 differentially expressed miRNAs (57 upregulated and 50 downregulated) were identified in TNBC cells compared with non-TNBC cells by using miRNA microarray analysis [48]. Yang et al. found that the circAGFG1 acts as a sponge of miR-195-5p to promote TNBC progression through regulating cyclin E1 (CCNE1) expression and to serve as a new diagnostic marker or target for treatment of TNBC patients [49]. Naorem et al. obtained four upregulated (miR-135b-5p, miR-18a-5p, miR-9-5p, miR-522-3p) and two downregulated (miR-190b, miR-449a) miRNAs, which were associated with the progression of TNBC and high diagnostic sensitivity and specificity of discriminating TNBC and non-TNBC samples via the meta-analysis approach [50]. Hu et al. showed miR-93 expression level in TNBC tissues was significantly upregulated compared to that in non-TNBC tissues [51].

Taken together, the dysregulated expression of miRNAs may serve as a potential vital tool for identifying critical biomarkers in TNBC patients. Additionally, there are masses of other dysregulated miRNAs, acting as oncogenes or tumor suppressors, which have been demonstrated to be involved in the epigenetic alteration, EMT, cancer stem cell (CSC) maintenance, invasion, metastasis, proliferation, apoptosis, prognostic, chemotherapy and radiotherapy resistance, and clinic therapy in TNBC. The changes play an important role in modulating gene expression and almost all of the “hallmarks of cancer” relevant pathways. In this scenario, miRNAs might represent not only the potential markers to discriminate TNBC from other breast cancer groups, but also the easily detectable biomarkers to participate in carcinogenesis, predicting prognosis and response to therapy, and epigenetic control.

## 4. MiRNAs in the Carcinogenesis of TNBC

Since miRNAs are an important class of gene expression regulators, as shown in Figure 2, they have important roles in epigenetic modulation and have a dual role in tumor-promoting activities (commonly termed as oncogenic miRNAs or oncomiRs) and tumor-suppressing in TNBC. OncomiRs mainly contribute to the inhibition of endogenous tumor suppressor genes of TNBC, whereas tumor-suppressive miRNAs target oncogenes. Altered miRNA expression is associated with epigenetic regulation, cell survival, and metastasis through regulation in EMT, stemness, proliferation, differentiation, apoptosis, autophagy, and many other biological pathways in TNBC [52].

### 4.1. MiRNAs in Epigenetic Mechanisms in TNBC

Epigenetic TNBCs are characterized by extensive low level DNA methylation, which leads to increased genome-wide instability. A great deal of key proteins associated with the biogenesis of miRNAs can be methylated and influence the production of transcribed miRNAs, or the aberrant methylation changes in the promoter CpG island region affect the number of transcribed miRNAs. Moreover, the miRNAs can target the DNA methyltransferase (DNMT) or histone deactylases (HDACs) to affect the DNA methylation or histone acetylation process to alter their expression.

MiRNAs were the targets of aberrant epigenetic modulation in TNBC (Table 2). The aberrant epigenetic regulation, especially the DNA hypermethylation of the promoter CpG island region, led to the inactivation of many tumor suppressor genes. A large proportion of miRNAs are located within the intronic regions of protein-coding genes, so they can be co-regulated. Nevertheless, some miRNA gene loci on the genome harbor their own promoter CpG island region, giving strong bases for their regulation by methylation-mediated silencing, with the characteristic of tumor suppressor genes. For instance, miR-31 is transcribed from the first intron of a novel long non-coding (lnc)RNA, LOC554202 on human chromosome 9 and the regulation of its transcriptional activity is under control of LOC554202. Augoff et al. found a major CpG island that spans the first exon of LOC554202, which was upstream of the miR-31 locus, suggesting an epigenetic regulation can be occurred by methylation of both miR-31 and its host gene. The downregulation expression of both miR-31 and the host gene LOC554202 in the TNBC cell lines is most probably provoked by a hypermethylation of its promoter-associated CpG island, resulting in TNBC progression [53]. A number of studies have shown that suppression of miR-200 family members is epigenetically regulated via the DNA methylation of the promoter in TNBC. Pang et al. showed that MYC proto-oncogene, bHLH transcription factor (MYC) physically interacted and recruited DNMT3A to the promoter of miR-200b, leading to proximal CpG island hypermethylation and subsequent miR-200b inhibition. Meanwhile, miR-200b directly inhibited the expression of DNMT3A, indicating a feedback loop mechanism between miR-200b and epigenetic molecules. The downregulated miR-200b can synergistically enhance target genes, such as zinc finger E-box binding homeobox 1 (ZEB1), SRY-box transcription factor 2 (SOX2), and CD133, to inhibit the capacities of migration, invasion, and mammosphere formation of TNBC cells [54]. Wee et al. also reported that different patterns of methylation in the miR-200b cluster promoter in different BC sub-types were associated with metastasis or hormone receptor status [55]. The low level expression of miR-200c was associated with the high levels of miR-200c/miR-141 locus methylation, which led to lymph node metastasis. The miR-200c/miR-141 locus methylation turned out to be significantly associated with high-ZEB1 expressing tumors, and the decreased expression of ZEB1 could reduce the DNA methylation of miR-200c/miR-141 locus and decrease histone H3K9 trimethylation, leading to cellular plasticity [56]. Taube et al. found that the expression of miR-203 is downregulated via epigenetic regulation (promoter DNA methylation) in less differentiated and mesenchymal-appearing EMT/CSC-enriched claudin-low TNBC cell lines compared to more differentiated, epithelial-appearing, luminal cancer cell lines [57]. The CpG methylation of miR-34a promoter was increased in breast cancer, leading to transcriptional silencing, which is dominant even in the presence of normal transactivation by p53 after DNA damage [58]. A report by the Lehmann group showed that miR-9-1, miR-124a3, miR-148, miR-152, and miR-663 were epigenetically inactivated through aberrant hypermethylation of their promoters. Notably that of miR-9-1, the reduction of its methylation and the concomitant reactivation of expression were observed after treatment with demethylating agent 5-aza-2-deoxycytidine (5-AZA) in breast cancer cell lines, but no unequivocal upregulation could be demonstrated in the other four miRNA genes [59]. Hoffman et al. indicated that the hypermethylation of a CpG rich region upstream of miR-196a-2 precursor is associated with reduced breast cancer risk [60].

MiRNAs are also the regulators of aberrant epigenetic modulation in TNBC (Table 2). DNA methylation is mostly catalyzed by three major forms of DNMTs: DNMT1 (maintenance DNMT), and DNMT3a and DNMT3b (de novo DNMTs). The histone acetyl transferases catalyze the acetylation progress, while HDACs catalyze the removal of acetyl groups. MiRNAs can control the epigenetic machinery directly or indirectly via regulating the expression of DNMTs, histone methyltransferases (HMTs), or ten–eleven translocations (TETs), termed Epi-miRNAs, in TNBC. For instance, Fabbri et al. identified that the expression of the miR-29 family is inversely correlated with DNMT3A/B in lung cancer tissues directly by intriguing complementarities to the 3′-UTRs, which is associated with poor prognosis. Interestingly, tumor suppressor genes, such as fragile histidine triad diadenosine triphosphatase (FHIT) and WW domain containing oxidoreductase (WWOX), are epigenetically-silenced (methylation-silenced) by promoter hypermethylation, and re-expressed as a consequence of their promoter CpG island demethylation, leading to cancer cell apoptosis, growth suppression, and inhibition of tumorigenicity [61]. The expression of DNMT3B is also under the control of miR-148 by binding to an unusual site in the coding sequence region (not the 3′-UTR), resulting in decreased DNA methylation levels, and it is responsible for the several different splice variants of DNMT3B (DNMT3B1, DNMT3B2, and DNMT3B4, but not DNMT3B3) [62]. On the other hand, miR-148a is also epigenetically regulated through its promoter hypermethylated from a self-amplified epigenetic feedback loop [59,63]. Another study demonstrated that the ectopic expression of miR-143 can directly target the DNMT3A, resulting in reducing hypermethylated DNA on phosphatase and tensin homolog (PTEN) gene promoter, and it increases TNF receptor superfamily member 10c (TNFRSF10C) methylation in TNBC cells [64]. Besides DNMT protein levels being modulated by miRNAs, the expression of HDAC protein is also regulated by miRNAs. The miR-200 can directly target the 3′-UTR of HDAC4 mRNA and inhibit the expression of HDAC4. In turn, HDAC proteins epigenetically downregulate miR-200 expression, suggesting an epigenetic feedback loop between HDAC4 and miR-200 [65]. Tryndyak et al. showed that miR-200 family-mediated transcriptional upregulation of E-cadherin in TNBC cell lines directly associated with the translational repression of ZEB1 and indirectly increased acetylation of histone H3 at the E-cadherin promoter, resulting in disruption of the repressive complexes between ZEB1 and HDAC, and the inhibition of sirtuin-1 (SIRT1). All of these influence the mesenchymal phenotype and drug-resistant phenotypes in TNBC cell lines [66]. In summary, these studies offer a better understanding of the interaction between epigenetic mechanisms–miRNAs and methylation or acetylation, which improve the knowledge of TNBC development, progression, diagnostics, and prognostics.

### 4.2. MiRNAs in EMT in TNBC

EMT is a key step in local invasion and distant metastasis in the initial event of cancer cells, as it allows the tumor cells to lose the expression of genes that promote epithelial phenotype, such as cell–cell contact, and gain the expression of mesenchymal markers to acquire more fibroblast-like features, and change the polarization of the epithelial cells and cytoskeleton to invade adjacent tissues [67]. EMT cascade is triggered by diverse extracellular and intracellular signals, including transforming growth factor, beta (TGF-β), Wnt/β-catenin, hypoxia inducible factor 1/2 (HIF1/2), Notch, nuclear factor kappa B (NF-κB), and Ras-extracellular regulated protein kinases 1/2 (ERK1/2), via EMT-related transcription factors, such as Src, Ras, Ets, integrins, snail family transcriptional repressor 1/2 (Snail1/2), SLUG, twist family bHLH transcription factor 1 (Twist1), ZEB1/2, SRY-box transcription factor 9 (Sox9), E47, and kruppel-like factor 8 (KLF8) [52].

In TNBC, an increasing number of miRNAs acting as oncogenes are involved in EMT (Table 3). A high level of miR-9 expression in TNBC patients is associated with EMT and breast cancer stem cell (BCSC) phenotypes, which directly target cadherin 1 (CDH1) to activate β-catenin and vascular endothelial growth factor (VEGF), leading to increased cell motility, invasiveness and tumor angiogenesis [68,69]. An investigation by Stinson et al. showed that miR-221/222 decreased the expression of epithelial-specific genes and increased the expression of mesenchymal-specific genes, which are essential for EMT regulation. As demonstrated in this research, miR-221/222 was stimulated directly by the transcription factor FOS-like 1 (FOSL1, also known as Fra-1), decreased the inhibition of mitogen-activated or extracellular signal-regulated protein kinase (MEK) in the RAS pathway, and repressed the expression of ZEB2 to downregulate E-cadherin, through targeting the 3′-UTR of trichorhinophalangeal syndrome type 1 (TRPS1) [70]. Hwang et al. found that depletion of another target of miR-221/222, adiponectin receptor 1 (ADIPOR1), activated the canonical NF-κB and subsequent NF-kB, interleukin 6 (IL6), and janus kinase 2 (JAK2)/signal transducer and activator of transcription 3 (STAT3) signaling axis, and induced EMT and increased cell invasion [71]. Recent data described oncogenic miR-181a as overexpressed in more aggressive breast cancers, which is particularly highly expressed in TNBC. Up-modulation of miR-181a enhanced TGF-β-mediated EMT and invasive phenotypes by suppressing the expression of proapoptotic Bim [72]. MiR-155 mediated the loss of CCAAT-enhancer binding protein beta (C/EBPβ) and promoted TNBC progression by shifting the TGF-β response from growth inhibition to EMT, invasion, and metastasis [73]. Additionally, another oncogenic miR-10b targeting TGF-β1 is upregulated in TNBC cell lines. The inhibition of its expression can increase E-cadherin expression and decrease vimentin expression, partially reversing EMT, invasion, and proliferation induced by TGF-β1 in breast cancer cells [74]. Additionally, the miR-103/107 reduces the expression of Dicer, the key component of the miRNA processing machinery, to promote cell migration and invasion in vitro. At the cellular level, miR-103/107 promotes EMT in breast cancer, attained by downregulating miR-200 levels, which targets the E-cadherin negative regulators ZEB1/2 [75,76].

In contrast, various oncosuppressive miRNAs are linked with EMT properties in TNBC (Table 3). For instance, the miR-200 family, which is comprised of miR-200a/b/c, miR-141, and miR-429, is a well-studied tumor-suppressive regulator in differentiated epithelial cells associated with EMT. The expression of all family members was obviously lower in metastatic TNBC in comparison to other types of breast cancer [77]. The expression of these miRNAs (except miR-141) was downregulated significantly in TGF-β1-induced EMT murine mammary epithelial cells. Overexpression of the miR-200 family miRNAs hindered EMT by enhancing E-cadherin expression through direct targeting of E-cadherin transcriptional repressors ZEB1 (also known as δEF1) and ZEB2 (also known as SIP1). Consistently, ectopic expression of the miR-200 family miRNAs significantly increased E-cadherin level and reversed cell morphology into epithelial phenotype in the 4TO7 mouse TNBC cell line [78]. Furthermore, a report by Gregory et al. supported the above conclusion. They reported that the expression of the miR-200 family was lost in invasive breast cancer cell lines with mesenchymal phenotype [79]. In detail, Howe et al. reported that miR-200c maintains the epithelial phenotype not only by targeting ZEB1/2, but also by inhibiting a great deal of gene, such as fibronectin 1 (FN1), moesin (MSN), neurotrophic tyrosine receptor kinase type 2 (NTRK2 or TrkB), leptin receptor (LEPR), and Rho GTPase activating protein 19 (ARHGAP19). Inhibition of MSN and/or FN1 is especially sufficient to mediate miR-200c to suppress cell migration [80]. Downregulation of miR-200b can increase the ability of EMT in TNBC cells by targeting ZEB1/2 and suppressing protein kinase Cα (PKCα) [74,81]. Recently, Soung et al. indicated that miR-200b was a major target gene of arrestin domain containing 3 (ARRDC3) and played an essential role in mediating ARRDC3 dependent reversal of EMT phenotypes in TNBC cells, suggesting a positive feedback loop between the two molecules [82]. MiR-141 was down-modulated in TNBC cells and found to have a positive role in epithelial phenotype maintenance of breast cancer [47]. Concomitant with the miR-200 family, the negative regulator miR-205 was dramatically reduced in TGF-β-induced cells to undergo EMT via direct targeting of ZEB1/2 [79]. MiR-199a-5p was observed downregulated in TNBC, and overexpression of miR-199a-5p in TNBC cells significantly altered EMT-related genes expression, such as CDH1, ZEB1, and TWIST by targeting of phosphatidylinositol-4,5-bisphosphate 3-kinase catalytic subunit delta (PIK3CD), reducing cell motility and invasiveness, as well as repressed tumor cell growth [83]. Similarly, miR-3178 was significantly reduced in TNBC, and the ectopic overexpression of miR-3178 suppressed cell proliferation, invasion, and migration by inhibiting the EMT by targeting of notch receptor 1 (Notch-1) in TNBC [84]. MiR-212-5p and miR-655 were downregulated in TNBC and associated with EMT phenotype by targeting paired related homeobox 1/2 (Prrx1/2) [85,86]. Recently, a study revealed that the miR-199/214 miRNAs cluster is underexpressed in TNBC cells, and upregulation of the miR-199/214 cluster decreases TNBC phenotype via its control of proliferation and EMT by specifically controlling the deposition of the collagen 1 (Col1) of extracellular matrix (ECM) protein components [25].

### 4.3. MiRNAs in Maintenance of Stemness in TNBC

CSCs or tumor-initiating cells (TICs) represent a small subpopulation of the tumor cells that possess the self-renewal and tumor-initiating capabilities through differentiation into the heterogeneous progeny of tumor cells. The CSCs displayed stem cell properties with the cell surface markers epithelial specific antigen (ESA^+^) and CD44 (CD44^+^), in the absence of marker CD24 (CD24^−/low^) [87]. The high proportion of CSCs in TNBC patients correlated with poor prognosis and recurrence, and possibly more resistance to chemo- and radiotherapy [88]. Studies have confirmed a direct link between the EMT and acquisition of stem cell-like properties [89].

Several oncogenic miRNAs have shown to be involved in CSC regulation for TNBC (Table 4). The upregulation of miR-21 targeting PTEN in TNBC correlated with the poor prognosis [90]. Han et al. demonstrated that antagonism of miR-21 reversed EMT phenotype with down-expression of hypoxia inducible factor 1 subunit alpha (HIF-1α), and suppression of migration and invasion breast cancer CSC (BCSC)-like cells [74]. Moreover, TGF-β, which seemed to induce a stem cells phenotype through increasing the population of breast cancer cells to induce sphere formation in suspension, stimulation upregulated the expression of miR-21 in cancer cells to increase HIF-1α expression and EMT process induced the BCSC-like phenotype [51]. Also, the overexpression of miR-181 family members, which comprise miR-181a, -b, -c, and -d, in TNBC is positively regulated by TGF-β induced the stemness and sphere formation in suspension at the post-transcriptional level by targeting the serine/threonine kinase ATM, suggesting that the TGF-β pathway and the miR-181 family interaction played key roles in regulating the CSC phenotype [72,91]. Another oncogenic miRNA, miR-495, is directly modulated by transcription factor E12/E47 and upregulated in CD44^+^/CD24^−/low^ and PROCR^+^/ESA^+^ (PROCR and short for protein C receptor) TNBC stem cells, suggesting it might be important for maintaining stem cell-like features, regulating in promoting cell invasion, enhancing cell proliferation in hypoxia through directly suppressed the expression of E-cadherin, and regulating in development and DNA damage responses-1 (REDD1) [92,93]. Li et al. showed that miR-221/222 was overexpressed in highly aggressive TNBC cells and promoted BCSC properties and tumor growth through downregulating PTEN, which in turn sustaining the Akt/NF-κB/cyclooxygenase-2 (COX-2) activation [94]. Cuiffo et al. reported that miR-199a enhances BCSC properties converge on and represses the expression of the transcription factor forkhead box p2 (FOXP2), and promotes breast cancer metastasis associated significantly with poor survival [95]. Additionally, aberrantly expressed oncogenic miR-20a downregulated the expression of MHC class I polypeptide-related sequence A/B (MICA/B), two ligands for the stimulatory natural killer (NK) cell receptor NKG2D, which enhanced the resistance of BCSC to NK cell cytotoxicity and promote lung metastasis [96].

Various miRNAs have oncosuppressive properties through multiple mechanisms involved in BCSCs in TNBC (Table 4), especially the miR-200 family and the let-7 family. The MiR-200 family is down-modulated in stem cells and essential for the maintenance of BCSC formation and growth; overexpression of miR-200c strongly suppresses the ability of normal mammary stem cells to form mammary ducts and tumor formation through the downregulation of polycomb ring finger oncogene B lymphoma Mo-MLV insertion region 1 homolog (Bmi-1) [97]. MiR-200b is associated with the increase of CSC formation, as well as the increase in Suz12 expression, which is the subunit of polycomb repressor complex 2 (PRC2) and the direct target of miR-200b [98]. Rokavec et al. indicated that monocyte-derived monocyte chemotactic protein-1 (MCP-1) induced the transformation of immortal breast epithelial cells, triggered by transient activation of MEK/ERK, IKK/NF-κB, and IL6 signaling, and maintained the transformed state by constitutive activation of a feed-forward inflammatory signaling circuit that consisted of the miR-200c, p65, mitogen-activated protein kinase 9 (Mapk9/JNK2), heat shock transcription factor 1 (HSF1) and IL6. Mechanically, IL6-mediated suppression of miR-200c activates the expression of JNK2 and p65/RelA, and then, the JNK2-dependent activation of HSF1 promotes IL6 expression via facilitating demethylation of the IL6 promoter to facilitate the binding of p65 and c-Jun and constitutive IL6 transcription. This signaling circuit drives in epithelial cell transformation and mammary cell tumorigenesis [99]. In addition, the let-7 family is a well-known tumor suppressor function on metastasis and stemness of TNBC cells. A report by the group of Iliopoulos showed that the transient activation of Src oncoprotein causes an epigenetic switch from immortalized breast cells to a transformed cell of self-renewing mammospheres containing cancer stem cells via triggering the NF-κB-mediated inflammatory response, which directly activates an inhibitor of miRNA processing, Lin28B, and reduces the expression of let-7. Then, the let-7 inhibits expression of IL6, both through directly binding its 3′-UTR and indirectly interacting with Ras, leading to suppression of the NF-κB activity. On the other hand, IL6 inhibition of let-7 expression occurs through NF-κB, suggesting there is a negative feedback loop between let-7a and IL6, controlled by NF-κB, which is required for maintenance of the transformed phenotype and stem cell population [100]. Yu et al. showed that let-7 is decreased in breast tumor-initiating cells (BT-ICs) and reduced let-7 is required to maintain mammospheres, proliferation but inhibits differentiation by targeting H-RAS and high mobility group AT-hook 2 (HMGA2) to regulate multiple stem cell-like properties of BT-ICs. That is, let-7 inhibits self-renewal but had no effect on differentiation in part by regulating H-RAS, while silencing HMGA2 causes differentiation but did not affect self-renewal [101]. Similar to let-7, Yu et al. demonstrated that miR-30 was inversely correlated with its target genes, ubiquitin-conjugating enzyme 9 (Ubc9) and integrin b3 (ITGB3), in the stem-like features of breast tumor-initiating cells (BT-ICs). Enforcing the expression of miR-30 inhibited the self-renewal of BT-ICs by reducing Ubc9, and triggered apoptosis in BT-ICs through ITGB3 and Ubc9 upregulation [102]. Guo et al. showed suppression of STAT3 activation or ectopic expression of let-7 and miR-200 reversed the mesenchymal phenotype of breast cancer cells, and was involved in cytokine-mediated reprogramming of self-renewal and differentiation in CSCs through the Lin28-let-7-HMGA2 and miR-200-ZEB1 signaling pathways [103]. Sun et al. indicated that let-7d sensitized TNBC stem cells to radiation-induced self-renewal repression through inhibition of the cyclin D1/Akt1/Wnt1 signaling pathway [104]. Aside from the miR-200 and let-7 families, the miR-15/16 (miR-16, miR-15b) and miR-103/107 (miR-103, miR-107) families, as well as miR-145, miR-335, and miR-128b, also inhibit CSC growth and are downregulated in CSCs by regulating common target genes of Bmi-1 and Suz12, as well as the DNA-binding transcription factors ZEB1/2 and kruppel-like factor 4 (Klf4), to form an inverse relationship between the levels of these miRNAs and their respective targets in TNBC [105]. MiR-203 was critical in both stemness and EMT in TNBC; ectopic miR-203 induced reversal of EMT, mesenchymal-to-epithelial transition (MET), and the forfeiture of self-renew capacity, resulting in suppressing proliferation and colony formation of TNBC cells via suppression of the primary TP63 isoform in mammary epithelia (ΔNp63α), which were robustly expressed in mammary stem cells (MaSCs) and vital to the maintenance of self-renew capacity in diverse epithelial structures [106]. MiR-205 was downregulated in TNBC carcinoma cells to promote EMT, interrupting epithelial cell polarity and expanding mammary stem cell and tumor stem cell populations by targeting Notch-2. Furthermore, spontaneously developed mammary lesions in miR-205-deficient mice markedly diminished breast cancer cell stemness through activation of miR-205 [107]. Additionally, increased miR-137 suppressed stemness significantly and decreased follistatin-like 1 (FSTL1) expression in TNBC cells via integrin β3/Wnt signaling or perturbing BAF chromatin remodeling complex subunit BCL11A (BCL11A)-DNMT1 interaction [108,109]. MiR-223 overexpression sensitized TNBC stem cells to tumor necrosis factor-related apoptosis-inducing ligand (TRAIL)-induced apoptosis by targeting HCLS1-associated protein X-1 (HAX-1) [110]. MiR-4319 suppressed the self-renewal and malignancy in TNBC stem cells through E2F transcription factor 2 (E2F2) [111]. Huang et al. found that IMP U3 small nucleolar ribonucleoprotein 3 (IMP3) plays a vital role in maintaining stem cell properties and promotes TNBC stemness through miRNA-34a regulation [112]. Additionally, by targeting SET domain bifurcated histone lysine methyltransferase 1 (SETDB1), miR-7 inhibited cell invasion and metastasis, decreased the BCSC population, and partially reversed EMT in TNBC cells by downregulating the STAT3 pathway [113]. Lin et al. reported miR-33b as a negative regulator of cell stemness and metastasis in breast cancer by targeting HMGA2, spalt-like transcription factor 4 (SALL4), and Twist1 [114].

### 4.4. MiRNAs in Regulation of Metastasis in TNBC

Metastasis is a complex multistep process in which tumor cells break away from the original site and migrate to other parts of the body to form metastatic lesions and new tumors. The metastatic ability of the primary breast tumor is a vital feature to estimate tumor grade and stage, and the treatment methods by pathological characterization. Oncogenic miRNAs play a vital role in cell migration, invasion, and metastasis (Table 5). Expression of miR-182 promotes cell proliferation, invasion, and migration via negatively regulate profilin 1 (PFN1) or forkhead-box F2 (FOXF2) [115,116]. Chen et al. reported that miR-10b and miR-373 were upregulated in lymph node metastasis breast cancer patients [117], especially, miR-373 promotes the EMT transition and metastasis via the thioredoxin-interacting protein (TXNIP)-HIF1-α-Twist signaling axis in breast cancer [118]. MiR-10b was reported to be induced by Twist to inhibit translation of homeobox D10 (HOXD10), resulting in increased expression of Ras homolog family member C (RHOC), invasiveness, and metastasis [119]. Elevated miR-21 expression may facilitate lymph node metastasis inversely associated with PTEN expression [120]. Avery-Kiejda found that 71 miRNAs were differentially expressed in TNBC, including the members of miR-200 family and the miR-17/92 oncogenic cluster, 27 miRNAs of which were associated with lymph node metastases from biological functions and pathways analysis [121]. Jin et al. also indicated that the metastatic potential of TNBC is decreased via caloric restriction-mediated reduction of the miR-17/92 cluster by targeting four potential genes related to extracellular matrix (ECM): Collagen 4 alpha 3 (COL4A3), laminin alpha 3 (LAMA3), and TIMP metallopeptidase inhibitor 2 and 3 (TIMP2/3) [122]. MiR-629-3p was identified as a risk factor for lung metastasis from TNBC through the regulation of a well-known metastatic suppressive gene, leukemia inhibitory factor receptor (LIFR) [123], while miR-455-3p promotes tumor cell proliferation, invasion, and migration by targeting etoposide induced 2.4 (EI24) in TNBC [124]. MiR-125b was also highly expressed in human TNBC tissues and cell lines, and inhibited the proliferation and metastasis by binding 3′-UTR of adenomatous polyposis coli (APC) and suppressed the activity of the intracellular Wnt/β-catenin pathways and EMT [125]. Furthermore, the miR-181a was involved in TNBC metastasis via elevated expression of Bim [72].

Oncosuppressive miRNAs also participate in the regulation of metastasis in TNBC (Table 5). MiR-200 family [77,126,127], miR-205 and miR-200c, were downregulated in tumor tissues and significantly associated with lymph node metastasis in TNBC [128]. A report by Aceto et al. indicated that Src-homology 2 domain-containing phosphatase 2 (SHP2) activated the ERKs, causing upregulation of ZEB1, as well as v-myc myelocytomatosis viral oncogene homolog (c-Myc), which resulted in the repression of let-7 miRNA and increased the expression of let-7 targets, including RAS and c-Myc, forming a key positive feedback signaling loop involved in the TNBC progression [129]. Ma et al. showed that large intergenic ncRNA-regulator of reprogramming (lincRNA-ROR) functions as a competing endogenous RNA (ceRNA) that sponges miR-145 and upregulates the expression of mucin1 (MUC1) to promote invasion and metastasis in TNBC [130]. MiR-145 was shown to regulate invasion and metastasis in TNBC by targeting small GTPase ADP-ribosylation factor 6 (Arf6) to affect E-cadherin localization and impact cell–cell adhesion [131]. Recent studies indicate that upregulation of miR-145 significantly reduces cell motility and invasiveness in TNBC cells by targeting 3′-UTR of junctional adhesion molecule A (JAM-A) and fascin through the effect on the expression of podocalyxin (PODXL), serpin E1 (SERPINE1), gamma-actin, transgelin, and myosin light chain 9 (MYL9) [132]. Loss of function of miR-206 is related to increased metastasis potential by targeting 3′-UTR of coronin, actin-binding protein, 1C (CORO1C) [133]. Similarly, the low expression of miR-30a was associated with high histological grade and more lymph node metastasis by targeting receptor tyrosine kinase like orphan receptor 1 (ROR1) [134]. MiR-190a and miR-940 were also significantly reduced in TNBC tissues to prevent metastasis and cell invasion [19,135]. Additionally, several miRNAs function as oncosuppressor miRNAs involved in metastasis of TNBC, such as miR-33b [114], miR-146a-5p [136], miR-150 [137], miR-124 [138] and miR-148a [139], miR-126-3p [140], miR-508-3p [141], miR-613 [142], miR-519d-3p [143], miR-26a [144], and miR-130a [145], by targeting HMGA2, SALL4 and Twist1, SRY-box transcription factor 5 (SOX5), HMGA2, ZEB2, wnt family member 1 (WNT1) and neuropilin 1 (NRP1), the regulator of G-protein signaling 3 (RGS3), ZEB1, disheveled-associated activator of morphogenesis 1 (Daam1), LIM domain kinase 1 (LIMK1), metadherin (MTDH), FOSL1, and zonula occludens 1 (ZO-1, also called TJP1), respectively.

### 4.5. MiRNAs in Cell Proliferation in TNBC

Uncontrolled growth and proliferation play a vital role in the TNBC. Currently, increasing evidences suggest that many miRNAs promote cell proliferation in TNBC at all stages as oncogenes (Table 6). For instance, miR-182 is obviously overexpressed in TNBC tissues, and inhibition of its expression significantly decreases the cell proliferation by negatively regulating PFN1 [115]. Upregulated miR-21 can promote TNBC cell proliferation in vitro by targeting the mRNA 3′-UTR of PTEN associated with poor prognosis [90]. MiR-21 also coordinates with miR-206 to co-target the two repressors, RAS p21 protein activator 1 (RASA1) and sprouty-related EVH1 domain-containing 1 (SPRED1), of RAS–ERK signaling to promote proliferation of TNBC cells [47]. Overexpression of miR-146a, miR-146b-5p, and miR-498 in TNBC cells negatively regulates BRCA1 expression to act on proliferation [146,147]. Besides, other panels of upregulated miRNAs, such as miR-20a-5p, miR-25-3p, miR-135b, and miR-502, are associated with the proliferation of TNBC and cell lines by targeting different genes. The expression of B-cell translocation gene 2 (BTG2) is negatively regulated by miR-25-3p, and indirectly activates the AKT and ERK-mitogen-activated protein kinase (MAPK) signaling pathways to mediate the proliferation of TNBC cell [148]. MiR-20a-5p promotes TNBC cell proliferation by targeting the Runt-related transcription factor 3 (RUNX3), as well as its direct downstream targets, Bim and p21 [149]. High expression of miR-135b in TNBC tissue and cells can promote proliferation and metastasis by targeting the 3′-UTR of APC and significantly suppressing APC expression [150]. In addition, Cantini and co-workers obtained that cluster miR-532/502 regulated the cell proliferation and the cell cycle transition from G to M phases, and miR-502 directly targeted the expression of H4K20 methyltransferase SET8 involved in cell proliferation and cell cycle [25]. 

In addition to oncogene roles, miRNAs can also play suppressive roles in TNBC (Table 6). For instance, miR-203 presents low expression in TNBC cell lines to suppress cell proliferation and migration by inhibiting baculoviral IAP repeat containing 5 (BIRC5) and LIM and SH3 protein 1 (LASP1) [151]. Wu et al. reported that the ectopic expression of miR-205 significantly inhibits cell proliferation and anchorage independent growth, as well as cell invasion, by binding the 3′-UTR of erb-b2 receptor tyrosine kinase 3 (ErbB3) and vascular endothelial growth factor A (VEGF-A). MiR-205 is also a novel transcriptional target of p53 that inhibits cell proliferation involved in cell cycle progression, clonogenic potential in vitro and tumor growth in vivo, at least partially by targeting the two newly identified genes, E2F transcription factor 1 (E2F1) and laminin subunit gamma 1 (LAMC1) [152,153]. Ren et al. showed that the overexpression of miR-200c inhibited the proliferation of TNBC cells and induced apoptosis in vitro by targeting the expression of a protein known as X-linked inhibitor of apoptosis (XIAP) [126]. MiR-34a, a potent endogenous tumor suppressor regulated by p53 network at the transcriptional level, is often significantly downregulated in TNBC. Deng et al. indicated that intracellular recovery of miR-34a inhibits breast cancer cell growth and migration via targeting the Notch-1 signaling pathway [154]. Moreover, the decreased expression of miR-34a is inversely correlated with the increased V-Erb-B2 avian erythroblastic leukemia viral oncogene homolog 2 (ErbB2) levels in breast cancer associated with the promotion of cell proliferation [155]. Furthermore, Adams et al. found that miR-34a was lost in TNBC and regulation of the expression of miR-34a in cell lines can inhibit proliferation and invasion, activate senescence, and promote sensitivity to dasatinib by targeting the proto-oncogene c-SRC, suggesting a negative feedback exists between miR-34a and c-SRC [156]. Another two miRNAs, miR-940 and miR-211-5p, are downregulated in TNBC associated with cell proliferation and migration. The expression of zinc finger protein 24 (ZNF24) is negatively regulated by miR-940 [157], while decreased expression of miR-211-5p is inversely correlated with the increased expression of SET binding protein 1 (SETBP1) in TNBC [158]. Overexpression of miR-1301 suppresses TNBC cell proliferation, migration, and colony formation in vivo and in vitro by binding to the 3′-UTR of enhancer of zeste 2 polycomb repressive complex 2 subunit (EZH2) [159]. Additionally, some other miRNAs with tumor suppressor properties include miR-146a-5p [136], miR-26a [144], miR-490-3p [160], miR-143-3p [161], miR-17-5p [162], miR-539 [163], miR-125b [164], miR-217 [165], and miR-589 [166], which are downregulated and involved in suppressing tumor cell proliferation in TNBC by targeting different oncogenes.

### 4.6. MiRNAs in Regulation of Apoptosis in TNBC

Apoptosis is a well-known process of programmed cell death (PCD) in organism development as a homeostasis mechanism to maintain the cell population and as a defense mechanism to remove infected cells by noxious agents or disease. Insufficient apoptosis often leads to uncontrolled cell proliferation, followed by tumourigenesis [167]. Some miRNAs function as oncogenes inhibiting tumor cell apoptosis in TNBC (Table 7). MiR-21 and miR-182, mentioned above, inhibit apoptosis in TNBC [90,115,168]. Song et al. showed that miR-301b was upregulated in TNBC specimens and cell lines, and inhibited the apoptosis induced by 5-FU by direct binding to 3′-UTR of CYLD lysine 63 deubiquitinase (CYLD) mRNA to activate NF-κB p65 [169]. A report by Wang et al. indicated that the overexpression of miR-155-5p promoted proliferation and reduced bufalin-induced apoptosis in TNBC cells by regulating forkhead box O3A (FOXO3A) [170]. Li et al. revealed that urokinase-type plasminogen activator receptor (uPAR) inhibits apoptosis through suppression of the c-myc-miR-17-5p/20a-death receptors 4/5 (DR4/DR5) pathway in TNBC [171]. Additionally, downregulation of miR-429 in two TNBC cell lines partially rescues the δ-tocotrienol-induced apoptosis in TNBC cells by targeting XIAP [172].

Oncosuppressive miRNAs are downregulated in TNBC to promote apoptosis through various signaling pathways, as listed in Table 7. MiR-200c induces apoptosis by targeting the expression of XIAP [126]. MiR-31 inhibits the oncogenic NF-κB pathway, and induces apoptosis, increasing sensitivity of chemo- and radiosensitivity in TNBC cell lines by direct inhibition of a positive regulator of B-cell CLL/lymphoma 2 (Bcl-2), protein kinase C epsilon (PKCε encoded by the PRKCE gene) [173]. Liu et al. reported that miR-4458 inhibited the cell proliferation and promoted cell apoptosis through negatively regulating the expression of suppressor of cytokine signaling 1 (SOCS1) [174]. Moreover, miR-890 inhibits cell proliferation and invasion, inducing apoptosis by negatively regulating its target gene CD147 in TNBC cells [175], and Zhang et al. showed that overexpression of miR-509 increased apoptosis and inhibited invasion probably by suppressing tumor necrosis factor-α (TNF-α) in TNBC cells [176]. Additionally, there are other panels of miRNAs with tumor suppressor properties involved in TNBC. For instance, miR-145 promotes TNF-α-induced apoptosis and facilitates the formation of receptor-interacting protein 1 (RIP1)-Fas-associated death domain (FADD)-caspase-8 complex by targeting cellular inhibitor of apoptosis (cIAP1) in TNBC cells [177]. Chen et al. unraveled the tumor-suppressive role of miR-199a-5p in TNBC associated with proliferation inhibition, cell cycle arrest, and increasing apoptosis by regulating the downstream targets of TGF-β2 and PIK3CD [83], and the overexpression of a novel tumor suppressor of miR-1296 involved in suppressed cell proliferation, cell cycle arrest, accompanied by induction of apoptosis by targeting cyclin D1 (CCND1) in TNBC cells [178]. Ke et al. showed that miR-10a inhibits proliferation and migration, promotes apoptosis via inhibiting phosphatidylinositol-4,5-bisphosphate 3-kinase catalytic subunit α (PIK3CA) expression at the post-transcriptional level to suppress phosphatidylinositol 3-kinase (PI3K)/protein kinase B (Akt)/mammalian target of rapamycin (mTOR) signaling, and induces the mitochondrial apoptotic pathway [179].

## 5. MiRNAs in Prognostic and Therapeutic Resistance in TNBC

The guidelines for treatment of most TNBC patients involve surgery combined with chemotherapy and radiotherapy (individually or in combination), while TNBC patients encounter poor prognosis and chemoresistance. Hence, accurately predicting prognostic would benefit TNBC treatment with chemotherapy agents after surgical resection.

### 5.1. Prognostic Implications of miRNAs in TNBC

With the increasing understanding of tumor biology, several studies have indicated that many miRNAs are associated with patients’ overall survival (OS), outcome, and recurrence, which are identified as the potential prognostic markers in TNBC. Jang et al. reported that high expression of miR-9 is closely related to survival and distant metastasis in TNBC patients [180]. Song et al. have shown that patients with low expression of lncRNA NEF and high expression of miR-155 in TNBC have a poor prognosis [181]. Kong et al. also showed that patients with high expression of miR-155 had a worse prognosis by targeting von Hippel-Lindau tumor suppressor (VHL) [182]. Toyama et al. found that miR-210 was highly expressed in Japanese TNBC patients, as a relatively independent factor to predict the OS for TNBC patients [183]. Yao et al. explored the expression of miR-493 in breast cancer samples using tissue microarrays (TMAs), and showed that patients with high expression of miR-493 had better disease-free survival [42,184]. Son et al. showed that miR-374a-5p was upregulated in TNBC patients. MiR-374a-5p was able to target arrestin beta 1 (ARRB1) that was downregulated in TNBC patients and the ARRB1 expression was inversely correlated with the histological grade of breast cancer and correlated with the survival of TNBC patients [185]. Tormo et al. showed that miR-449 affects prognosis, mainly by regulating the response of TNBC to the chemotherapy drug doxorubicin. The high expression of miR-449a was significantly associated with a good prognosis by Metabric study-GEO database, while miR-449b/c was not associated with prognosis [186]. Tsiakou et al. analyzed the single nucleotide polymorphisms (SNPs) of miR-34, and the result showed that the TNBC patients with TC/CC genotype in the rs4938723 C>T polymorphism had shorter OS intervals with worse prognosis [187]. In summary, these studies have identified the application of various miRNAs that can serve as a potential prognostic marker for TNBC patients (Table 8).

### 5.2. The Therapeutic Resistance of miRNAs in TNBC Therapy

The development of chemoresistance and radiation resistance remains a significant obstacle in TNBC treatment. The resistance is often multifactorial, and involves the maintenance of CSCs, the increase in capacity of repairing DNA damage, and the trigger of apoptosis by miRNAs (Table 9). Studies have revealed that the upregulation of miR-5195-3p, miR-18a, and miR-1207-5p is a potential predictor of TNBC sensitivity to paclitaxel or Taxol [188,189,190]. Wu et al. obtained that overexpression of miR-620 could promote the resistance of TNBC patients to gemcitabine by decreasing dCMP deaminase (DCTD) expression [191]. Li et al. found that abnormal expression of miR-770 can inhibit the resistance of TNBC cells to doxorubicin, mainly through regulation of apoptosis and tumor microenvironment [192]. Moreover, Rizzo et al. showed the expression of miR-15b, miR-23a, miR-26a, miR-29a, miR-106b, miR-128, miR-149, miR-181a, miR-192, miR-193b, miR-195, miR-324-3p, and miR-494 in TNBC cells treated with doxorubicin under short-term starvation (STS) condition were deregulated in drug sensitivity/drug resistance-related pathways [193]. Tang et al. demonstrated that taurine upregulated 1 (TUG1) has a sponge effect on miR-197, and TUG1 inactivates the WNT signaling pathway by modulating miR-197/nemo-like kinase (NLK), thereby increasing the sensitivity of TNBC to cisplatin [194]. Wang et al. found tamoxifen reversed EMT, and inhibited cell migration by demethylation of the miR-200c promoter in TNBC cells [195]. Li et al. showed that high levels of miR-105 and miR-93-3p were associated with poor survival in TNBC patients via Wnt/β-catenin signaling by downregulating secreted frizzled related protein 1 (SFRP1) to promote stemness, chemoresistance, and metastasis in TNBC cells [196]. Additionally, miR-21 links EMT and inflammatory signals (IL-6/STAT3/NF-κB-mediated signaling loop and activating the PI3K pathway) to confer resistance to trastuzumab and chemotherapy in HER2-positive breast cancer patients through downregulation of PTEN and programmed cell death 4 (PDCD4) [197].

Similar to chemotherapy, radiotherapy sensitivity also shows to be associated with miRNA levels in TNBC treatment (Table 9). For instance, increased expression of miR-95 in human breast cancer specimens indicates the resistance to radiation treatment by targeting sphingosine-1-phosphate phosphatase 1 (SGPP1), an antagonist of sphingosine-1-phosphate signaling in TNBC cells [198]. Liang et al. found that the expression of miR-302 was decreased in irradiated breast cancer and inversely correlated with AKT1 and RAD52. The downregulated expression of miR-302 confers radioresistance in TNBC cells in vitro and vivo, and replacement therapy sensitizes TNBC cells to radiotherapy [199]. A study by Ren et al. exposed that the expression of miR-27a is significantly higher, and miR-27a modulates proliferation and radiosensitivity by directly targeting cell division cycle 27 (CDC27) in TNBC cells [200]. Additionally, high miR-155 decreases the efficiency of homologous recombination DNA repair and enhances sensitivity to ionizing radiation (IR) in vitro and in vivo by directly targeting the 3′-UTR of RAD51, with better overall survival of TNBC patients [201]. On the other hand, multidrug resistance (MDR) is considered a major obstacle in the failure in treatment of TNBC with conventional chemotherapeutic and radiotherapy. For example, miR-638 overexpression increases sensitivity to DNA-damaging agents, ultraviolet (UV) and cisplatin, to reduce DNA repair capability, thereby reducing proliferation, invasive ability, and DNA repair capabilities by targeting BRCA1 in post UV/cisplatin-exposed TNBC cells [202]. Deng et al. indicated that overexpression of miR-143-3p reverses MDR by inhibiting the expression of its target protein cytokine-induced apoptosis inhibitor 1 (CIAPIN1) of TNBC in vivo [203].

## 6. Targeting miRNAs for TNBC Therapy

A great deal of oncogenic miRNA and tumor-suppressive miRNAs participate in cell proliferation, apoptosis, and metastasis associated with drug resistance. Both oncogenic miRNAs and tumor-suppressive miRNAs can be used as candidates for diagnoses and therapeutic purposes. There are two major approaches for developing miRNAs as anti-TNBC agents functioning as antagonist and mimic oligonucleotides. MiRNA antagonists, also called antagomirs or antimiRs, are single-stranded anti-miRNA oligonucleotides (AMOs) that are designed to directly complement mature miRNAs and acquire a gain of function in TNBC (gene-silencing therapy). MiRNA mimics, also known as miRNA replacement therapy, are used to restore a loss of function in malignant cells or to use miRNAs encoded in expression vectors (replacement therapy) [204].

### 6.1. MiRNA Antagonist Therapy in TNBC

MiRNA antagonist antisense oligonucleotides act as competitive inhibitors of miRNAs by being fully or partially complementary to the target miRNA strand, resulting in degradation or repression of functional miRNA by annealing to the miRNA target. One potential complication is that antagomir binds to unintended RNAs as off-target; therefore, the precise hybridization between the antagonist miRNA and the endogenous miRNA is critical. An antagomir to miR-10b significantly decreases miR-10b levels and increases the expression of miR-10b target gene, HOXD10. Meanwhile, the tumor metastasis was markedly suppressed in a mouse mammary tumor model [205]. A therapeutic RNA nanotechnology for specific and efficient delivery of anti-miR-21 (antagomiR-21) blocked the growth and triggered cancer cell apoptosis of TNBC in orthotopic mouse models by targeting mRNAs of PTEN and PDCD4 [206]. Yin et al. also indicated that delivery of anti-miR21 using RNA nanoparticles with high specific targets gained promising results for TNBC therapy [207]. Moreover, Devulapally et al. synthesized antisense-miR-21 and antisense-miR-10b loaded by PLGA-b-PEG polymer nanoparticles (NPs), which effectively blocked the endogenous miR-21 and miR-10b function, inhibited the proliferation and metastasis of TNBC cells as multi-target antagonization [208]. Zhou et al. used a novel calcium phosphate–polymer hybrid nanoparticle system to co-encapsulate and co-deliver a combination of therapeutic agents with different physicochemical properties, such as the inhibitors for miRNA-221/222 (miRi-221/222) and paclitaxel, which suppressed the proliferation and significantly enhanced the therapeutic efficacy of paclitaxel [209]. Furthermore, Ahir et al. showed that tailored-CuO-nanowire decorated with folic acid mediated coupling of the mitochondrial ROS generation and miR425–PTEN axis in inducing apoptosis and retarding migration to furnish potent anti-cancer activity in TNBC cells [210]. In addition, another approach is miRNA sponges, which are vector-encoded molecules using synthetic mRNAs; they contain strong promoters, multiple and tandem binding sites to multiple interested miRNAs simultaneously. These sponges can also be integrated into the genome to create stable cell lines or transgenic animals [211]. A study by Huang et al. indicated that miR-150 sponge inhibited growth and clonogenicity and induced apoptosis via increasing the binding of the P2X7 receptor in TNBC cells lines [212].

### 6.2. MiRNA Mimic (miRNA Replacement) Therapy in TNBC

MiRNA mimic approaches, as in the traditional gene therapy, also known as miRNA replacement therapy, provide a new opportunity to replace the unfunctional tumor suppressor genes through introduction of analogous miRNA molecules. The miRNA mimic was designed to easily bind to the silencing complex and precisely target the endogenous miRNA. MRX34, a liposome-formulated miR-34a mimic compound, the first clinical investigation of miRNA replacement agents in cancer treatment, was used in patients with advanced or metastatic hepatocellular carcinoma by intravenous injection. The results showed that MRX34 treatment with dexamethasone premedication has antitumor activity in some patients with refractory advanced solid tumors [213]. However, this clinical trial was terminated prematurely due to serious side effects [214]. A study by Di et al. indicated that either transient expression of miR-34a synthetic mimics or lentivirus-based miR-34a-stable enforced expression inhibited growth and triggered apoptosis in multiple myeloma (MM), synthetic miR-34a downregulated the target of Bcl-2, cyclin dependent kinase 6 (CDK6), and Notch-1 at both the mRNA and protein level, while lentivirus-based miR-34a-stable also inhibited tumor growth and improved survival in mice bearing TP53-mutated MM xenografts without systemic toxicity [215]. In TNBC, Deng et al. revealed that miR-34a was co-encapsulated with doxorubicin into hyaluronic acid (HA)–chitosan (CS) nanoparticles, and delivered into breast cancer cells synchronously to enhance antitumor effects by suppressing Bcl-2. The restoration of miR-34a inhibited cell migration via targeting Notch-1 signaling [154]. Recently, Wang et al. showed that a HA-decorated polyethylenimine–poly(d, l-lactide-co-glycolide) (PEI–PLGA) nanoparticle system was used for co-delivery of doxorubicin and miR-542-3p for TNBC treatment. The restoration of miR-542-3p triggered apoptosis through activating p53 and inhibiting survivin expression for TNBC therapy [216]. The recombinant adeno-associated viral vectors (rAAV) have now been used in cancer therapy due to their low pathogenicity and immunogenicity, as well as high transferring ability and long-term gene expression [217]. A study by Kota et al. disclosed that overexpression of miR-26a via AVV suppressed tumorigenesis by inhibition of cancer cell proliferation and induction of tumor-specific apoptosis without hepatoxicity or dysregulation of endogenous miRNA in a mouse model of hepatocellular carcinoma (HCC) [218]. Trepel et al. showed that a therapeutic suicide gene, delivered systemically by the dual-targeted AAV vector for the treatment of multifocal breast cancer, significantly inhibited tumor growth and overcame the side effects of collateral tropism [219].

### 6.3. Challenges and Perspectivse on miRNA Therapy in TNBC

The application of miRNAs as possible targets for therapeutic intervention against proliferation and metastasis provides an immense opportunity to develop better treatment for TNBC, with less toxicity and drug resistance. However, the major challenges of miRNA-based TNBC therapy are the design of effective miRNA mimics and suitable delivery systems without off-target. The low molecular weight and stability of miRNAs are the advantages [220]. Modified therapeutic miRNAs, such as the AMOs, cholesterol-conjugated antagomiRs, locked nucleic acid (LNA)-modified oligonucleotides and 2′-O-methoxyethyl-4′-thioRNA (MOE-SRNA) are used in clinic [35]. Meanwhile, new efficient delivery methods that prevent miRNA degradation and excretion are necessary. These delivery vehicles should be non-toxic with low immunogenicity, high efficiency, and highly accuracy [16]. Research focusing on latent delivery systems, including neutral lipid emulsion, polymer nanoparticles, solid lipid nanoparticles, liposomes and virus-based approaches, has been reviewed [221].

TNBC is a challenging cancer subtype, and miRNA therapy combined with chemotherapy might provide systemic treatment. Yang et al. indicated that the upregulation of miR-195 with mimic oligonucleotides in MCF-7/ADR cells (an Adriamycin-resistant MCF-7 subline) increased the sensitivity of breast cancer to Adriamycin treatment, reduced the tumor cell survival and promoted tumor cell apoptosis through inhibition of Raf-1, Bcl-2, and P-glycoprotein expression [222]. Gong et al. demonstrated that blocking the action of miR-21 with antisense oligonucleotides re-sensitized the resistance to trastuzumab therapy by inducing growth arrest, proliferation inhibition, and cell-cycle blocking via targeting of PTEN in TNBC cell lines [223]. Wang et al. showed that the hyaluronic acid-coated PEI–PLGA nanoparticles mediated co-delivery of doxorubicin and miR-542-3p increased both drug uptake and cytotoxicity in TNBC cells, which further promoted TNBC cell apoptosis via activating p53 and suppressing survivin expression [216]. In summary, the theoretical applications of miRNA therapy in TNBC are very promising.

## 7. Conclusions

TNBC is a very aggressive breast cancer type with poor prognosis and high cancer mortality due to lack of targeted medicines applied in this subtype. A growing body of evidence provides support that miRNAs act as oncogenes or tumor suppressors, involved in the cell proliferation, apoptosis, and metastasis in TNBC. MiRNAs might represent not only the potential markers for diagnosis, but also the precise targeted therapies. In this review, we summarized the recent advances related to miRNAs involved in carcinogenesis and progression in TNBC, and classified these miRNAs as promising biomarkers for prognosis and therapies for TNBC patients based on their regulation mechanisms through EMT, stemness, proliferation, and apoptosis. We also noticed that one miRNA can influence multiple genes, thus some miRNAs have multiple roles in diagnosis, prognosis, and prediction of therapeutic response in TNBC, such as miR-200 family, miR-21 and miR-182. Thus, two major approaches of miRNA mimics and antagonists provide promising anti-TNBC therapies with challenges. On-going research progress in studying miRNA biology and miRNA delivery system innovations would lead to better understanding, diagnosis, and treatment of TNBC in the near future.

## Figures and Tables

**Figure 1 cells-08-01492-f001:**
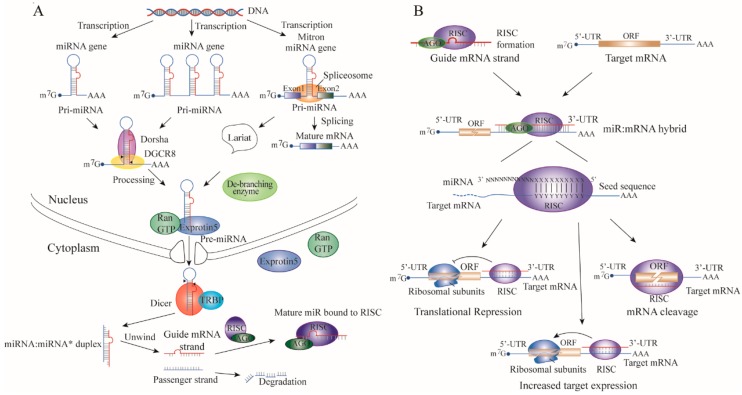
Steps involved in the biogenesis and functions of miRNAs. (**A**) The miRNAs’ synthesis and maturation pathway. MiRNAs are transcribed by RNA polymerase II enzyme as about 500–3000 nucleotides-long pri-miRNA capped with 7-methylguanosine at its 5′-end and polyadenylated tail at the 3′-end. The pri-miRNAs form a specific secondary structure of hairpin-shape and stem–loop, and are then cleaved by the RNase III family enzymes Drosha-DGCR8 based on the stem–loop structure, which cleaves 11 base pairs away from the single-stranded stem–loop junction to shorten the pri-miRNA to 60–70 nucleotides with double helix hairpin structure to yield the pre-microRNA that contains a 5′-phosphate group and 2–3 nucleotides overhang stretch at the 3′-end. Then, the pre-miRNA is exported to the cytoplasm by the shuttle protein Exportin-5 in an Ran-GTP-dependent manner, where the pre-miRNA is cleaved by the endonuclease cytoplasmic RNase III enzyme Dicer to yield an approximately 22 nucleotides dsRNA with two-nucleotide 3′-overhang miRNA:miRNA* duplex in concert with TRBP or PACT cofactors. The miRNA:miRNA* duplex unwinds by cytoplasmic helicase and one of the strands is defined as a mature miRNA, and miRNA* is quickly degraded. The mature miRNA, AGO2, and other proteins are incorporated into the RISC to form an miRISC. (**B**) The mechanisms of miRNA action. The activity of mature miRNAs binding to the target mRNA sequence occurs in a perfect or, most often, in an imperfect manner. The complementarity binding is usually restricted to the 5′-end nucleotides 2–8 of the miRNA, termed “seed sequence”, with the mRNA complementary sequences at 3′-UTR. Due to decreased steric hindrance, the perfect miRNA:mRNA complementarity usually lead to degradation of the mRNA by AGO2 via the small interfering RNA (siRNA) pathway and is referred to as “slicer activity”. The imperfect pairing with mRNA causes the RNA-Polymerase to become blocked and leads to the low-level efficiency of mRNA transcription at the initiation, or elongation and termination. In addition, the miRNAs can bind the promoter regions and the coding regions to upregulate the translation of their target mRNA, but this is in the minority.

**Figure 2 cells-08-01492-f002:**
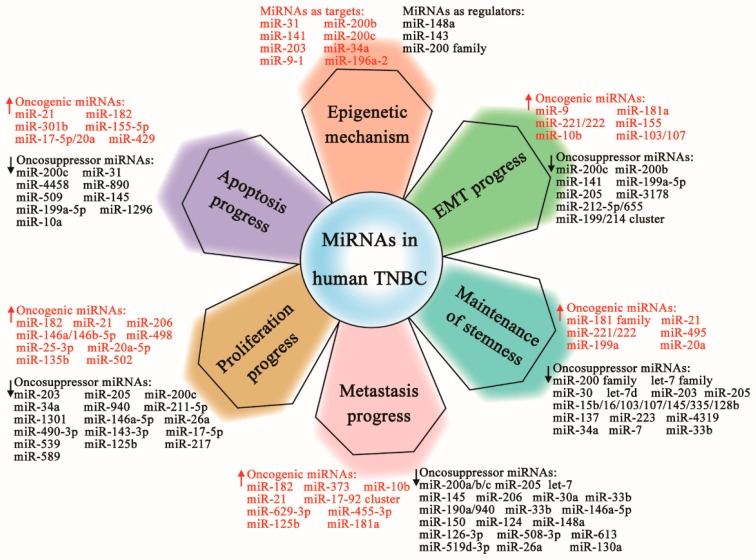
The multiple roles of miRNAs in TNBC. The miRNAs have a role as targets or regulators involved in aberrant epigenetic regulation in TNBC. Moreover, miRNAs function as oncogenic (Upregulated, 

) or oncosuppressive genes (Downregulated, 

) associated with epithelial–mesenchymal transition (EMT), stemness maintenance, invasion and metastases, cell proliferation and survival, and apoptotic regulation via special targets, resulting in the relapse, development, or inhibition of TNBC.

**Table 1 cells-08-01492-t001:** The dysregulation of microRNAs in triple-negative breast cancer (TNBC).

miRNA	Type of Deregulation	Samples Type	Reference
**Differentially Expressed miRNAs Between TNBC and other breast cancer (BC) Subtypes**			
miR-17-92 cluster	Upregulated	Tumor tissues	[40]
miR-205, miR-342	Downregulated	Tumor samples	[38,41]
miR-155, miR-493, miR-30e, miR-27a	Deregulated	Tumor samples	[42]
miR-17-5p, miR-18a-5p, miR-20a-5	Upregulated	Tumor samples	[43]
miR-21, miR-221, miR-210, let-7a	Upregulated	Tumor samples	[44]
miR-195, miR-145	Downregulated	Tumor samples	[44]
**Differentially Expressed miRNAs Between TNBC and Non-TNBC**			
miR-21, miR-146a, miR-182, miR-10b	Upregulated	Tumor samples	[45]
miR-190a, miR-136-5p, miR-126-5p, miR-135b-5p, miR-182-5p	Downregulated	Serum	[46]
miR-146a, miR-100, miR-125b, miR-29a, miR-222, miR-221, miR-138-5p, miR-4324, miR-4800-3p, miR-6836-3p	Upregulated	TNBC cell lines	[47]
miR-200a, miR-200c, miR-141, miR-375, miR-203, miR-363-5p, miR-182-5p, miR-141-3p, miR-339-5p, miR-4655-3p, miR-4784, miR-664b-5p, miR-6787-5p	Downregulated	TNBC cell lines	[47]
miR-190a, miR-136-5p, miR-126-5p	Downregulated	Tumor samples	[19]
miR-135b-5p, miR-182-5p	Upregulated	Tumor samples	[19]
miR-195-5p	Downregulated	Tumor samples	[49]
miR-135b-5p, miR-18a-5p, miR-9-5p, miR-522-3p	Upregulated	Tumor samples	[50]
miR-190b, miR-449a	Downregulated	Tumor samples	[50]
miR-93	Upregulated	Tumor samples	[51]

**Table 2 cells-08-01492-t002:** MiRNAs associated with epigenetic mechanisms in TNBC.

MiRNA	Direct/Indirect Targets	Functions	Reference
**MiRNAs as the Targets of Aberrant Epigenetic Modulation**			
miR-31	WAVE3, RhoA, Radexin, integrin	Metastasis suppressor	[53]
miR-200b	ZEB1, SOX2, CD133	Inhibit the capacities of migration, invasion, and mammosphere formation	[54,55]
miR-200c/141	ZEB1	Lymph node metastasis, cellular plasticity	[56]
miR-203	DKK1	EMT and cancer stem cell properties	[57]
miR-34a	CDK6	Tumor suppressor gene	[58]
miR-9-1	Unknown	Associated with the early and frequent event in breast cancer development	[59]
miR-196a-2	HOXB2, HOXB3, HOXC13, HOXB5	A delay in the G2/M phase of the cell cycle	[60]
**MiRNAs as the Regulators of Aberrant Epigenetic Modulation**			
miR-148a	DNMT3B	Associated with the primary human breast cancer development	[59,63]
miR-143	DNMT3A	Inhibits proliferation and soft agar colony formation, downregulates the expression of DNMT3A	[64]
miR-200 family	HDAC4	Influences the mesenchymal phenotype and drug-resistant phenotypes	[65,66]

**Table 3 cells-08-01492-t003:** MiRNAs associated with EMT in TNBC.

MiRNA	Direct/Indirect Targets	Functions	Reference
**Oncogenic miRNAs**			
miR-9	CDH1	Increases cell motility and invasiveness, tumor angiogenesis	[68,69]
miR-221/222	ZEB2, TRPS1, ADIPOR1	Induce EMT and increase cell invasion, activate the NF-kB	[70,71]
miR-181a	Bim	Enhances TGF-β-mediated EMT, migratory, and invasive phenotypes	[72]
miR-155	C/EBPβ	Promotes EMT, invasion, and metastasis	[73]
miR-10b	TGF-β1	Promotes EMT, invasion, and proliferation	[74]
miR-103/107	Dicer	Promotes migration, invasion, and EMT	[75,76]
**Oncosuppressor miRNAs**			
miR-200 family	ZEB1/2	Reverse EMT	[77,78,79]
miR-200c	ZEB1/2, FN1, MSN, NTRK2, LEPR, ARHGAP19	Maintain the epithelial phenotype, suppress cell migration	[80]
miR-200b	ZEB1/2, PKCα	Reverse EMT phenotypes	[74,81,82]
miR-141	Unknown	Epithelial phenotype maintenance	[47]
miR-205	ZEB1/2	Reduce TGF-β-induced EMT	[79]
miR-199a-5p	PIK3CD	Alters EMT-related genes expression, reduces cell motility and invasiveness, represses tumor cell growth	[83]
miR-3178	Notch-1	Inhibits EMT, suppress proliferation, invasion, and migration	[84]
miR-212-5p/655	Prrx1/2	Inhibits EMT phenotype	[85,86]
miR-199/214 cluster	Col1	Decreases TNBC phenotype via its control of proliferation and EMT	[25]

**Table 4 cells-08-01492-t004:** MiRNAs associated with maintenance of stemness in TNBC.

MiRNA	Direct/Indirect Targets	Functions	Reference
**Oncogenic miRNAs**			
miR-21	PTEN	Increases EMT process, induces breast cancer stem cell (BCSC)-like phenotype, promotes migration and invasion	[51,74,90]
miR-181 family	ATM	Induces a stem cell phenotype	[72,91]
miR-495	E-cadherin, REDD1	Maintain stem cell-like features, promote cell invasion and proliferation in hypoxia	[92,93]
miR-221/222	PTEN	Promotes BCSC properties and tumor growth	[94]
miR-199a	FOXP2	Enhances BCSC properties, promotes metastasis	[95]
miR-20a	NKG2D, MICA/B	Promote the lung metastasis by enhancement of BCSC resistance to NK cell cytotoxicity	[96]
**Oncosuppressor miRNAs**			
miR-200 family	Bmi-1, Suz12,	Regulate BCSC formation and growth	[97,98,99]
let-7 family	H-RAS, HMGA2	Regulate the mammosphere formation, BCSC self-renewal and metastasis	[100,101]
miR-30	Ubc9, ITGB3	Inhibit the self-renewal of breast tumor-initiating cells (BT-ICs), trigger apoptosis	[102]
let-7d	Cyclin D1	Induces stem cells radiation sensitization	[104]
miR-15b/16/103/107/145/335/128b	Bmi-1, Suz12, ZEB1/2, Klf4	Inhibit cancer stem cell (CSC) growth	[105]
miR-203	ΔNp63α	Forfeiture of self-renewing capacity associated with epithelial stem cells, suppresses proliferation and colony formation	[106]
miR-205	Notch-2	Inhibits EMT and stem cell properties	[107]
miR-137	FSTL1	Suppresses TNBC stemness	[108,109]
miR-223	HAX-1	Re-sensitizes TNBC stem cells to tumor necrosis factor-related apoptosis	[110]
miR-4319	E2F2	Suppresses the self-renewal and malignancy in stem cells	[111]
miR-34a	IMP3	Regulates TNBC stem cell property	[112]
miR-7	SETDB1	Inhibits cell invasion and metastasis, decreases the BCSC population, and partially reverses EMT	[113]
miR-33b	HMGA2, SALL4, Twist1	Regulate cell stemness and metastasis	[114]

**Table 5 cells-08-01492-t005:** MiRNAs associated with metastasis in TNBC.

MiRNA	Direct/Indirect Targets	Functions	Reference
**Oncogenic miRNAs**			
miR-182	PFN1, FOXF2	Promote cell proliferation, invasion, and migration	[115,116]
miR-373	TXNIP	Induces cancer cell EMT and metastasis	[117,118]
miR-10b	HOXD10	Promotes invasion and metastasis	[119]
miR-21	PTEN	Lymph node metastasis	[120]
miR-17/92 cluster	COL4A3, LAMA3, TIMP2/3	Lymph node metastases, promote metastasis	[121,122]
miR-629-3p	LIFR	Lung metastases	[123]
miR-455-3p	EI24	Promotes proliferation, invasion, and migration	[124]
miR-125b	APC	Promotes proliferation, metastasis, and EMT	[125]
miR-181a	Bim	Promotes EMT, migratory, and invasive	[72]
**Oncosuppressor miRNAs**			
miR-200a/b/c	PKCα, UBASH3B, XIAP	Suppress proliferation, migration, invasion, and metastasis, promote apoptosis	[77,126,127]
miR-205	Unknown	Lymph node metastasis	[128]
let-7	RAS, c-Myc	Block growth and reduce metastasis	[129]
miR-145	MUC1, Arf6, JAM-A, Fascin	Reduce cell motility, invasiveness, and metastasis	[130,131,132]
miR-206	CORO1C	Regulates metastasis	[133]
miR-30a	ROR1	Associates with high histological grade and more lymph node metastasis	[134]
miR-190a/940	Unknown	Prevents metastasis and cell invasion	[19,135]
miR-33b	HMGA2, SALL4, Twist1	Inhibit metastasis	[114]
miR-146a-5p	SOX5	Inhibits proliferation and metastasis	[136]
miR-150	HMGA2	Inhibits metastasis	[137]
miR-124	ZEB2	Inhibits invasion and metastasis	[138]
miR-148a	WNT1, NRP1	Suppress metastasis	[139]
miR-126-3p	RGS3	Inhibits proliferation, migration, invasion, and angiogenesis	[140]
miR-508-3p	ZEB1	Inhibits cell invasion and EMT	[141]
miR-613	Daam1	Inhibits cell migration and invasion	[142]
miR-519d-3p	LIMK1	Suppresses growth and motility	[143]
miR-26a	MTDH	Suppresses proliferation and metastasis	[144]
miR-130a	FOSL1, ZO-1	Suppress migration and invasion	[145]

**Table 6 cells-08-01492-t006:** MiRNAs associated with proliferation in TNBC.

miRNA	Direct/Indirect Targets	Functions	Reference
**Oncogenic miRNAs**			
miR-182	PFN1	Promotes TNBC cell proliferation	[115]
miR-21	PTEN	Promotes TNBC cell proliferation	[90]
miR-206	RASA1, SPRED1	Promote TNBC cell proliferation	[47]
miR-146a/146b-5p	BRCA1	Increases BRCA1-mediated proliferation	[146]
miR-498	BRCA1	Promotes TNBC cell proliferation	[147]
miR-25-3p	BTG2	Promotes TNBC cell proliferation	[148]
miR-20a-5p	RUNX3, Bim, p21	Promote TNBC cell proliferation	[149]
miR-135b	APC	Promotes proliferation and metastasis	[150]
miR-502	STE8	Promotes cell proliferation and cell cycle	[25]
**Oncosuppressor miRNAs**			
miR-203	BIRC5, LASP1	Suppress cell proliferation and migration	[151]
miR-205	ErbB3, VEGF-A, E2F1, LAMC1	Inhibit cell proliferation, cell invasion, cell cycle arrest, clonogenic potential	[152,153]
miR-200c	XIAP	Inhibits proliferation, induces apoptosis	[126]
miR-34a	Notch-1, ErbB2, c-SRC	Inhibit cell growth and invasion, activate senescence, sensitize to dasatinib	[154,155,156]
miR-940	ZNF24	Inhibits cell proliferation and migration	[157]
miR-211-5p	SETBP1	Inhibits cell proliferation and migration	[158]
miR-1301	EZH2	Suppresses proliferation, migration, and colony formation	[159]
miR-146a-5p	SOX5	Inhibits the proliferation and metastasis	[136]
miR-26a	MTDH	Suppresses tumor proliferation and metastasis	[144]
miR-490-3p	TNKS2	Inhibits the growth and invasiveness	[160]
miR-143-3p	LIMK1	Suppresses the growth	[161]
miR-17-5p	ETV1	Suppresses cell proliferation and invasion	[162]
miR-539	LAMA4	Inhibits proliferation and migration	[163]
miR-125b	MAP2K7	Inhibits proliferation	[164]
miR-217	KLF5	Inhibits cell growth, migration	[165]
miR-589	MTA2	Decreases cell proliferation, migration, and invasion	[166]

**Table 7 cells-08-01492-t007:** MiRNAs associated with apoptosis in TNBC.

MiRNA	Direct/Indirect Targets	Functions	Reference
**Oncogenic miRNAs**			
miR-21	PTEN	Promotes the tumor proliferation and inhibits cell apoptosis	[90]
miR-182	PFN1, RIP1	Promote the tumor proliferation, inhibit cell apoptosis and migration	[115,168]
miR-301b	CYLD	Inhibits cell apoptosis induced by 5-FU	[169]
miR-155-5p	FOXO3A	Promotes proliferation and reduces bufalin-induced apoptosis	[170]
miR-17-5p/20a	DR4/DR5	Inhibit apoptosis	[171]
miR-429	XIAP	Regulates apoptosis	[172]
**Oncosuppressor miRNAs**			
miR-200c	XIAP	Induces apoptosis	[126]
miR-31	Bcl-2, PKCε	Induce apoptosis, increase sensitivity of chemo- and radiosensitivity	[173]
miR-4458	SOCS1	Inhibits proliferation and promotes apoptosis	[174]
miR-890	CD147	Inhibits the cell proliferation and invasion, induces apoptosis	[175]
miR-509	TNF-α	Increases apoptosis and inhibits invasion	[176]
miR-145	cIAP1	Promotes TNF-α-induced apoptosis	[177]
miR-199a-5p	TGF-β2, PIK3CD	Proliferate inhibition, cell cycle arrest, and increase apoptosis	[83]
miR-1296	CCND1	Suppresses proliferation, cell cycle arrest accompanied by induction of apoptosis	[178]
miR-10a	PIK3CA	Inhibits the proliferation and migration, promotes apoptosis	[179]

**Table 8 cells-08-01492-t008:** MiRNAs associated with prognosis in TNBC.

MiRNA	Targets	Functions	Prognostic	Reference
miR-9	CDH1	Invasion and metastasis	Highly expressed with poor disease-free and distant metastasis-free survival	[180]
miR-155	VHL	Migration and invasion, tumor angiogenesis	Highly expressed with poor prognosis	[181,182]
miR-210	ISCU1/2, SDHD	Mitochondrial dysfunction	Highly expressed with worse disease-free and overall survival	[183]
miR-493	Unknown	Subclassify core basal and five negative phenotype subtypes	Highly expressed with better disease-free survival	[42,184]
miR-374a-5p	ARRB1	Promotes cell survival, proliferation, and migration	Highly expressed with poor prognosis	[185]
miR-449	CDK2, CCNE2	Doxorubicin resistance	Elevated expression with better survival in chemotherapy-treated	[186]
miR-34	GATA-X transcription factor	Tumor suppression	TC and CC alleles are associated with unfavorable prognosis	[187]

**Table 9 cells-08-01492-t009:** MiRNAs associated with therapeutic resistance in TNBC.

MiRNA	Targets	Functions	Drugs Involved	Reference
miR-5195-3p	EIF4A2	Enhances the chemosensitivity	Paclitaxel	[188]
miR-18a	Dicer	Increases PTX IC50 and reduces PTX-induced cell apoptosis	Paclitaxel	[189]
miR-1207-5p	LZTS1	Enhances cell growth arrest and cell apoptosis, a predictor of sensitivity towards Taxol	Taxol	[190]
miR-620	DCTD	Induces cell apoptosis and cell growth arrest, facilitates the resistance of gemcitabine	Gemcitabine	[191]
miR-770	STMN1	Regulates apoptosis and tumor microenvironment, suppresses the doxorubicin resistance and metastasis	Doxorubicin	[192]
miR-15b/23a/26a/29a/ 106b/128/149/181a/192 /193b/195/324-3p/494	Unknown	Mediates the anticancer effects of short-term starvation in doxorubicin-treated breast cancer cells	Doxorubicin	[193]
miR-197	NLK	Enhances cisplatin sensitivity, cell proliferation inhibition	Cisplatin	[194]
miR-200c	Vimentin, E-cadherin	Inhibit cell migration and enhances chemosensitivity of mesenchymal by reversing their EMT-like property	Tamoxifen	[195]
miR-105/93-3p	SFRP1	Promotes stemness, chemoresistance, and metastasis	Cisplatin	[196]
miR-21	PTEN, PDCD4	Sustain EMT and shape the tumor immune microenvironment, confer resistance to trastuzumab and chemotherapy	Trastuzumab	[197]
miR-95	SGPP1	Increass tumor growth and resistance to radiation treatment	Ionizing radiation	[198]
miR-302	AKT1, RAD52	Sensitize to radiotherapy	Ionizing radiation	[199]
miR-27a	CDC27	Modulates proliferation and radiosensitivity	Irradiation treatment	[200]
miR-155	RAD51	Decreases the efficiency of homologous recombination DNA repair and enhances sensitivity to ionizing radiation	Ionizing radiation	[201]
miR-638	BRCA1	Reduces proliferation, invasive ability, and DNA repair capabilities	UV/cisplatin	[202]
miR-143-3p	CIAPIN1	Effectively reverses multidrug resistance	Multidrug resistance	[203]

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
