# Peer review of "MicroRNAs Involved in Carcinogenesis, Prognosis, Therapeutic Resistance, and Applications in Human Triple-Negative Breast Cancer"

_cells, 2019, doi:10.3390/cells8121492_

Round 1
Reviewer 1 Report
The authors performed an in-depth analysis of the current state of the problem, and targeted microRNA targets for the diagnosis, prognosis and prediction of therapeutic response in triple-negative breast cancer. The article is written in good language, beautifully illustrated. I think that manuscript entitled "MicroRNAs Involved in the Carcinogenesis, Prognosis, Therapeutic Resistance and Applications in Human Triple-negative Breast Cancer" should be accepted in its current form for publication in Cells.
Author Response
Response to Reviewer 1 Comments
Point 1: Comments and Suggestions for Authors:
The authors performed an in-depth analysis of the current state of the problem, and targeted microRNA targets for the diagnosis, prognosis and prediction of therapeutic response in triple-negative breast cancer. The article is written in good language, beautifully illustrated. I think that manuscript entitled "MicroRNAs Involved in the Carcinogenesis, Prognosis, Therapeutic Resistance and Applications in Human Triple-negative Breast Cancer" should be accepted in its current form for publication in Cells.
Response 1: Thank you very much for your recognition of our manuscript.
Reviewer 2 Report
The authors summarized miRNAs involved in triple-negative breast cancer based on their regulation on EMT, stemness, proliferation, apoptosis and drug resistance. Also, the authors discussed about targeting miRNAs for triple-negative breast cancer therapy. This is a very informative review with comprehensive figures and tables. Only one minor comment:
It will be nice to include a table for the session 4.2 The therapeutic resistance of miRNAs in TNBC therapy, having miRNAs, targets, drugs involved….Author Response
Response to Reviewer 2 Comments
Point 1: Comments and Suggestions for Authors
The authors summarized miRNAs involved in triple-negative breast cancer based on their regulation on EMT, stemness, proliferation, apoptosis and drug resistance. Also, the authors discussed about targeting miRNAs for triple-negative breast cancer therapy. This is a very informative review with comprehensive figures and tables. Only one minor comment:
It will be nice to include a table for the session 4.2 The therapeutic resistance of miRNAs in TNBC therapy, having miRNAs, targets, drugs involved….
Response 1: We added table 9 (page 20) that summary the miRNAs associated with the therapeutic resistance in TNBC in the revised manuscript.
Reviewer 3 Report
The authors performed a comprehensive litarature review on microRNAs involved in tiple-negative breast cancer (TNBC).
Comments:
Key miRNAs that are differentially expressed between TNBC and other BC subtypes should be discussed and summarized in a table.The authors discuss a paper from Iliopoulos et al 2010 (ref. 84). They should also include a paper from the same group on this topic (Iliopoulos, 2009, Cell), where they showed a connection between NF-kappaB, Let-7, and IL-6 in breast cancer. Furthermore, the paper from Rokavec et al 2012 (Molelular Cell), which shows important findings on the link between IL-6 and miR-200c in BC, should be discussed in this review.
In the 4th section (prognosis): miRNAs that were associated with the prognosis of TNBC should be summarized in a table.
The authors mention that a liposome formulated miR-34a mimic compound was tested in a phase I clinical study and that this was the first clinical investigation of miRNA replacement agents in cancer treatment. It should be also mentioned that this clinical trial was terminated due to severe side effects.
English grammar should be extensively revised.
Author Response
Response to Reviewer 3 Comments
Thanks for your precious comments:
Point 1: Key miRNAs that are differentially expressed between TNBC and other BC subtypes should be discussed and summarized in a table.
Response 1: The key miRNAs that are differentially expressed between TNBC and other BC subtypes, TNBC and non-TNBC was discussed in section 3 (page 5 to 7, “miRNA dysregulation in TNBC”) and the associated miRNAs are summarized in table 1 (page 6 to 7) in the revised manuscript.
Point 2: The authors discuss a paper from Iliopoulos et al 2010 (ref. 84). They should also include a paper from the same group on this topic (Iliopoulos, 2009, Cell), where they showed a connection between NF-kappaB, Let-7, and IL-6 in breast cancer. Furthermore, the paper from Rokavec et al 2012 (Molecular Cell), which shows important findings on the link between IL-6 and miR-200c in BC, should be discussed in this review.
Response 2: We discussed the results of Iliopoulos et al 2009 (Cell) and Rokavec et al 2012 (Molecular Cell) (page 13) in the revised manuscript.
Point 3: In the 4th section (prognosis): miRNAs that were associated with the prognosis of TNBC should be summarized in a table.
Response 3: We added table 8 (page 19) that summary the miRNAs associated with prognosis in TNBC and table 9 (page 20) that summary the miRNAs associated with the therapeutic resistance in TNBC in the revised manuscript.
Point 4: The authors mention that a liposome formulated miR-34a mimic compound was tested in a phase I clinical study and that this was the first clinical investigation of miRNA replacement agents in cancer treatment. It should be also mentioned that this clinical trial was terminated due to severe side effects.
Response 4: We added the contents “However, this clinical trial was terminated prematurely due to serious side effects [214].” (page 22) in the revised manuscript.
Point 5: English grammar should be extensively revised.
Response 5: We would like extensively revise our English grammar through the language editing service offered by MDPI in the revised manuscript. We are happy to pay the service fee together with the publish charge.